# The Effect of Standardised Leaf Extracts of *Gaultheria procumbens* on Multiple Oxidants, Inflammation-Related Enzymes, and Pro-Oxidant and Pro-Inflammatory Functions of Human Neutrophils

**DOI:** 10.3390/molecules27103357

**Published:** 2022-05-23

**Authors:** Piotr Michel, Sebastian Granica, Karolina Rosińska, Małgorzata Glige, Jarosław Rojek, Łukasz Poraj, Monika Anna Olszewska

**Affiliations:** 1Department of Pharmacognosy, Faculty of Pharmacy, Medical University of Lodz, 1 Muszynskiego St., 90-151 Lodz, Poland; karolinaa.rosinskaa@gmail.com (K.R.); gosiaju11@gmail.com (M.G.); jaroslaw.rojek@o2.pl (J.R.); lukasz.poraj@gmail.com (Ł.P.); monika.olszewska@umed.lodz.pl (M.A.O.); 2Microbiota Lab, Centre for Preclinical Studies, Department of Pharmacognosy and Molecular Basis of Phytotherapy, Medical University of Warsaw, 1 Banacha St., 02-097 Warsaw, Poland; sgranica@wum.edu.pl

**Keywords:** *Gaultheria procumbens*, leaves, antioxidant activity, neutrophils, ROS, IL-1β, elastase-2

## Abstract

The leaves of *Gaultheria procumbens* are polyphenol-rich traditional medicines used to treat inflammation-related diseases. The present study aimed to optimise the solvent for the effective recovery of active leaf components through simple direct extraction and verify the biological effects of the selected extract in a model of human neutrophils ex vivo. The extracts were comprehensively standardised, and forty-one individual polyphenols, representing salicylates, catechins, procyanidins, phenolic acids, and flavonoids, were identified by UHPLC–PDA–ESI–MS^3^. The chosen methanol–water (75:25, *v*/*v*) extract (ME) was obtained with the highest extraction yield and total phenolic levels (397.9 mg/g extract’s dw), including 98.9 mg/g salicylates and 299.0 mg/g non-salicylate polyphenols. In biological tests, ME revealed a significant and dose-dependent ability to modulate pro-oxidant and pro-inflammatory functions of human neutrophils: it strongly reduced the ROS level and downregulated the release of pro-inflammatory cytokines and tissue remodelling enzymes, especially IL-1β and elastase 2, in cells stimulated by *f*MLP, LPS, or *f*MLP + cytochalasin B. The extracts were also potent direct scavengers of in vivo relevant oxidants (O_2_^•−^, ^•^OH, and H_2_O_2_) and inhibitors of pro-inflammatory enzymes (cyclooxygenase-2, hyaluronidase, and lipoxygenase). The statistically significant correlations between the tested variables revealed the synergic contribution of individual polyphenols to the observed effects and indicated them as useful active markers for the standardisation of the extract/plant material. Moreover, the safety of ME was confirmed in cytotoxicity tests. The obtained results might partially explain the ethnomedicinal application of *G. procumbens* leaves and support the usage of the standardised leaf extract in the adjuvant treatment of oxidative stress and inflammation-related chronic diseases.

## 1. Introduction

The genus *Gaultheria* Kalm ex L. (Ericaceae) contains over 150 species varying from low, ground-hugging shrubs to small trees native to Asia, Australasia, and North and South America [1,2]. The *Gaultheria* plants are known to accumulate a vast diversity of polyphenols [3,4], including methyl salicylate as the principal constituent of the essential oil [5,6], and non-volatile compounds: salicylate glycosides, procyanidins, and flavonoids [7,8,9]. Traditionally, plants abundant in polyphenols, especially salicylates, have been used worldwide as extracts, tinctures, infusions, and decoctions to treat several inflammation-related diseases cross-linked with oxidative stress [10]. Their popularity as adjuvant therapeutics is connected with their pleiotropic biological activity and better safety profile than synthetic drugs [11]. Therefore, they might serve as a good starting point for developing new, safe, and functional herbal medicinal products, including standardised extracts, which ensure an elevated concentration of active components and reproducible therapeutic effects [12]. Among various traditional plant materials, the leaves of evergreen plants, such as *Gaultheria* species, are especially industrially attractive due to their accessibility throughout the growing season, except in the winter months when the plants are covered with snow [1].

One of the most popular species in the *Gaultheria* genus is *Gaultheria procumbens* L. (American wintergreen, eastern teaberry), a small, low-growing shrub with evergreen leaves and red berry-like fruits, characterised by a distinctive wintergreen scent [1]. In ethnomedicine, the leaves, the fruits, or whole overground parts of the plant are used to treat inflammation-related disorders, especially rheumatoid arthritis, swelling, muscular pain, chronic tracheitis, and pharyngitis [7,13]. Although the plant is rich in essential oil with a potent antimicrobial activity [6], the main active components of its fruits, stems, and leaves are considered non-volatile polyphenols [8,9,14,15,16]. However, the chemical composition of the leaves is still insufficiently recognised, mainly due to the lack of available phenolic standards. Consequently, the standardisation of the leaf extracts is constricted.

The rational medical use of natural products required the verification of their biological effects [11,12]. Up to this point, the previous research on the biological activity of *G. procumbens* leaves has focused primarily on their antioxidant capacity [15,16], with scarce data available on their anti-inflammatory potential [15]. However, all earlier reports have been based only on simple, non-cellular in vitro methods, such as the DPPH, FRAP, and TBARS assays, and inhibition tests of pro-inflammatory enzymes–hyaluronidase (HYAL) and lipoxygenase (LOX) [15,16]. The accumulated results suggested that eastern teaberry leaves are a promising source of health-promoting polyphenols with potential medicinal applications, but further research is needed to confirm their biological effects in cell-based or in vivo studies. Moreover, as extraction conditions, especially extraction solvents, might significantly influence the composition and biological capacity of the *Gaultheria* extracts [8,9], solvent optimisation is also required.

Therefore, the present study aimed to select the best solvent for the extraction of *G. procumbens* leaf polyphenols and evaluate the impact of leaves on pro-inflammatory and pro-oxidant functions of human neutrophils ex vivo. The studies were performed on the dry extracts prepared by direct extraction of the plant material with four solvents, i.e., methanol–water (75:25, *v*/*v*), acetone, ethyl acetate, and *n*-butanol. The extracts were fully standardised by UHPLC–PDA–ESI–MS*^3^*, HPLC–PDA, and UV spectrophotometric assays using authentic phenolic standards isolated earlier from aerial parts of *G. procumbens* [14]. In addition, six non-cellular antioxidant capacity tests, such as the DPPH, TBARS, and FRAP assays, and the O_2_^•−^, ^•^OH, and H_2_O_2_ scavenging tests, and three pro-inflammatory enzymes inhibition assays towards HYAL, LOX, and cyclooxygenase 2 (COX-2), were used for solvent optimisation. Eventually, the selected extract was investigated for its effects on the pro-oxidant and pro-inflammatory functions of human neutrophils, including the release of reactive oxygen species (ROS), cytokines (IL-1β, IL-8, TNF-α), and tissue remodelling enzymes, such as matrix metalloproteinase 9 (MMP-9) and elastase-2 (ELA-2). Moreover, the cellular safety of the extract was tested by flow cytometry and propidium iodide staining.

## 2. Results

### 2.1. Phytochemical Standardisation of the Leaf Extracts

The qualitative phytochemical analysis of the investigated extracts was performed using the UHPLC–PDA–ESI–MS^3^ method. Forty-one phenolic compounds were detected, and thirty-eight were fully or tentatively identified (Figure 1, peaks **1–41**; Appendix A) based on their chromatographic (retention time) and spectral properties (UV–vis, MS/MS) compared with authentic standards and literature data [8,9,14,15,16,17]. The identified compounds belonged to five main polyphenolic classes, including simple phenolic acids (peaks **1**, **2**, **3**, **7**); quinic acid pseudodepsides (**5**–**7**, **9**, **10**, **12**, **19**, **34**), with dominating chlorogenic acid isomers (**5**, **10**, **12**); free catechins (**18**) and oligomeric procyanidins (**8**, **13–15**, **22–26**, **28**–**30**, **32**, **33**); methyl salicylate glycosides (**11**, **16**, **20**); and flavonoid glycosides (**31**, **35–39**) and aglycones (**41**). The number of compounds detected and identified in the tested extracts decreased as follows: methanol–water (75:25, *v*/*v*) extract (ME; 40 detected constituents, 37 fully or tentatively identified) > *n*-butanol extract (BE; 38, 35) > ethyl acetate extract (EAE; 27, 25).

The quantitative phenolic profile of the individual and total polyphenols of *G. procumbens* leaves was investigated spectrophotometrically and by the HPLC–PDA method, developed and fully validated previously [14]. As shown in Table 1, the extracts differed significantly in phenolic contents and extraction yields.

The total phenolic content of the extracts varied in a wide range for both Folin–Ciocalteu-reactive substances (TPC, 168.1–302.4 mg GAE/g dw) and low-molecular- weight constituents detectable by RP–HPLC (TPH, 204.6–336.7 mg/g dw), with the highest levels observed in ME and EAE, respectively (Table 1). Except for ME, the TPH levels of all extracts were significantly higher than their TPCs, which could be explained by the prevalent contribution of methyl salicylate glycosides in EAE and BE with their total levels (TSAL, 128.6–288.1 mg/g dw) constituting 85.6% and 62.9% of the TPH values, respectively. The main salicylate was methyl salicylate 2-*O*-*β*-d-xylopyranosyl-(1→6)-*β*-d-glucuronopyranoside (gaultherin, peak **20**, Figure 1) with a 99.4–100.0% contribution to the TSAL values. Interestingly, taking into account the extraction efficiency and expressing the results in mg per g of dry leaves (Figure 2), the ME showed a similar TSAL recovery from the leaf material (36.8 mg/g) compared to BE (38.0 mg/g) and six times higher than EAE (5.8 mg/g), despite the lower content in the extract.

Flavan-3-ols and procyanidins formed the second relevant group of leaf polyphenols. The total proanthocyanidin contents, determined by the *n*-butanol/HCl method (TPA), varied in the range of 36.9–174.4 mg/g (Table 1). They were two to four times higher than the total levels of low-molecular-weight procyanidins (TLPA, 9.7–63.9 mg/g, measured by HPLC), which indicated a significant contribution of highly polymerised homologs among the leaf proanthocyanidins. The most promising profile, i.e., high TPA level with the elevated relative contribution of TLPA with higher potential bioavailability, was revealed in ME, which was also the most abundant in (−)-epicatechin (peak **18**), procyanidin B2 (peak **14**), and cinnamtannin B1 (peak **24**) as individual components. In addition, ME yielded the highest TPA (64.8 mg/g) and TLPA (23.8 mg/g) recovery, expressed per dw of leaves (Figure 2), when compared to EAE and BE.

Flavonoids (TFL, 18.6–49.4 mg/g) formed the third fraction of the analysed extracts. Similarly to procyanidins, flavonoids were concentrated mainly in ME (Table 1). Moreover, the TFL recovery with ME (expressed per dw of leaves) was 2.5 and 48 times higher than in BE and EAE, respectively (Figure 2). The dominant glycoside miquelianin (quercetin 3-*O*-*β*-d-glucuronopyranoside, peak **37**) constituted 8.3–65.1% of the TFL fraction.

Minor components of the extracts were phenolic acids (TPHA), among which chlorogenic acid isomers prevailed as their total contents (TCHA) reached up to 67.5% of the TPHA values (Table 1). The highest TPHA content was in EAE. However, as shown in Figure 2, ME recovered the highest TPHA (4.2 mg/g), TCHA (2.5 mg/g) and SPHA (1.7 mg/g) amounts from the leaves.

### 2.2. Antioxidant and Anti-Inflammatory Activity in Non-Cellular In Vitro Models

The investigated extracts turned out to be potent scavengers of free radicals, both synthetic (DPPH) and generated in vivo (O_2_^•−^, ^•^OH, and H_2_O_2_), inhibitors of linoleic acid peroxidation (TBARS), and ferric ions reducers (FRAP), compared to antioxidant standards (Table 2). All extracts revealed a concentration-dependent activity in all applied models. The antioxidant capacity of the tested extracts decreased in the following order: ME > BE > EAE. The most effective antioxidant among all extracts was ME; its activity parameters did not differ significantly (*p* > 0.05) from those of BE only in the H_2_O_2_ scavenging test. In other tests, the capacity of ME was up to two to three times higher than that of BE and EAE.

Regarding the anti-inflammatory potential of the extracts, they revealed a significant and concentration-dependent ability to inhibit HYAL and COX-2, with a less pronounced response towards LOX, compared to synthetic anti-inflammatory drugs dexamethasone and indomethacin (Table 3). The most potent COX-2 inhibitor among the extracts was ME; it acted up to two times stronger than BE and EAE. The anti-inflammatory activity in HYAL and LOX tests decreased in the following order: BE > ME > EAE.

The activity and composition data matrix (Appendix A) revealed robust linear relationships between most tested variables. A substantial and statistically significant correlation was found between the antioxidant activity parameters and TPA (|r|≥ 0.76), TLPA (|r|≥ 0.90), and TFL (|r|≥ 0.62) levels, and between the IC_50_ values in the enzyme inhibition assays and TSAL (|r|≥ 0.93), TPA (|r|≥ 0.62), and TLPA (|r|≥ 0.82) values. The high correlation coefficients suggest that proanthocyanidins and flavonoids are primarily responsible for the antioxidant activity of the leaf extracts, while both salicylates and procyanidins contribute to their anti-inflammatory activity. Additionally, in the case of procyanidins, higher correlation coefficients observed for TLPA than for TPA indicate low-molecular-weight representatives as mainly responsible for the observed effects.

### 2.3. Influence on Pro-Oxidant and Pro-Inflammatory Functions of Human Neutrophils Ex Vivo

#### 2.3.1. Effects on Neutrophil Viability

The most promising extract in terms of polyphenol recovery and activity in non-cellular tests (ME) was subjected to a cell-based study in a model of human neutrophils. In the first stage, the impact of ME on cell viability was investigated by flow cytometry and propidium iodide (PI) staining. As shown in Figure 3, ME at 50–150 µg/mL did not cause a significant reduction in the cell membrane integrity compared to the control (*p* > 0.05). Eventually, the percentage of the PI(+) cells was 4.00 ± 0.74% and 3.49–5.95% for the ME-treated neutrophils and control cells, respectively, and the extract was considered non-cytotoxic within the tested concentration range.

#### 2.3.2. Effects on ROS Production

The cellular antioxidant activity of ME was evaluated by measuring its impact on ROS production by human neutrophils stimulated with *N*-formyl-l-methionyl-l-leucyl-l-phenylalanine (*f*MLP). The extract revealed significant (*p* < 0.05) and dose-dependent antioxidant effect within the whole concentration range of 50–150 µg/mL, equivalent to 20–60 µg TPH + TPA/mL (Figure 4A). For instance, at 150 µg/mL, it downregulated the ROS release by 94%, compared to the *f*MLP-stimulated neutrophils. Furthermore, at 100 µg/mL, ME reduced the ROS level to 21.5%. It allowed for reaching the physiological value of ROS generation, which in non-stimulated cells was 26.3%.

#### 2.3.3. Effects on the Release of Pro-Inflammatory Cytokines and Enzymes

As illustrated in Figure 4B–F, ME demonstrated a concentration-dependent ability to modulate the release of pro-inflammatory cytokines (IL-1β, IL-8, and TNF-α) and tissue remodelling enzymes (ELA-2 and MMP-9) produced by neutrophils after stimulation by bacterial lipopolysaccharide (LPS) or *f*MLP with cytochalasin B, depending on the test. The observed effects were the most potent for the levels of IL-1β and ELA-2. In the presence of ME at 150 µg/mL (equivalent to 60 µg TPH + TPA/mL)**,** the release of IL-1β decreased to 61.4% (*p* < 0.01) compared to the stimulated cells and was not significantly different from the effect of DEX at 25 µM (*p* > 0.05). Likewise, the ELA-2 secretion by the stimulated neutrophils decreased to 57.7% at 150 µg/mL of ME (*p* < 0.001). This value did not differ from that observed for the positive standard of quercetin at 25 µM (*p* > 0.05). In addition, ME at 100–150 µg/mL reduced the ELA-2 release to levels not different from the physiological values observed in the non-stimulated cells (77.1%).

The effects of ME on MMP-9, IL-8, and TNF-α secretion were less pronounced but still statistically significant at 100–150 µg/mL (*p* < 0.05). Moreover, in the case of MMP-9, the ME effect at 150 µg/mL was similar to that of DEX at 25 µM (*p* > 0.05).

## 3. Discussion

Medicinal plants have been used for centuries as traditional herbal remedies to treat numerous disorders. Nowadays, standardised plant extracts are preferred due to their elevated efficiency and reproducibility compared to the source plant materials [18]. The solvent selection is thus one of the most critical stages in the production of herbal medicines, as it determines the phytochemical composition and biological activity of the prepared extracts [19,20].

Therefore, the first step of the present work concerned solvent optimisation to maximise the recovery of bioactive constituents of *G. procumbens* leaves. For this reason, four solvents of different polarities, i.e., methanol–water (75:25, *v*/*v*), acetone, ethyl acetate, and *n*-butanol, were used for the plant extracts preparation. The use of hydroalcoholic solution enabled us to produce an extract (ME) corresponding to the herbal formulations (tinctures) most commonly used in traditional medicine [21], while acetone, ethyl acetate, and *n*-butanol were used for selective extraction/concentration of individual groups of polyphenols, according to our previous studies of eastern teaberry stems [8] and fruits [9]. However, the preliminary phytochemical profiling revealed that the acetone extract had a similar qualitative composition to ME with a lower total phenolic content and extraction yield (data not shown), which excluded acetone from further studies.

The previous research carried out by our team involved phytochemical and in vitro non-cellular biological activity testing of fractionated *G. procumbens* leaf extracts [15]. The fractionated extraction enabled the concentration of the phenolic fractions but was laborious, time-consuming, and difficult to apply on a larger scale to produce plant-based formulations. Therefore, in the present work, a fast and straightforward direct extraction method was applied.

The chemical composition of plant extracts may vary greatly depending on the extraction solvent, extraction procedure, and due to the seasonal variability of the plant material, which in turn is influenced by several climatic and geographical factors [22]. As chemical composition has a crucial impact on the biological activity of the extracts, an important stage in their studies is therefore their thorough phytochemical standardisation, which was also performed for the tested extracts.

As a result, and thanks to the previous isolation and complete structural determination of several compounds from the aerial parts of *G. procumbens* [14], it was possible to accurately characterise the qualitative profile of the leaves for the first time. Consequently, thirty-eight out of forty-one detected leaf polyphenols (Figure 1, Appendix A) were completely or tentatively identified. Moreover, several compounds, i.e., methyl salicylate. 2-*O*-*β*-d-glucopyranosyl-(1→2)-[*O*-*β*-d-xylopyranosyl-(1→6)]-*O*-*β*-d-glucopyranoside (peak **11**), a gaultherin isomer (**16**), cinnamtannin B-1 (**24**), procyanidin C1 (**25**), and 3-*O*-*β*-d-xylopyranosyl-(1→2)-*β*-d-glucuronopyranosides of quercetin (**31**) and kaempferol (**38**), were compared with the previously isolated standards of rare or new natural products [14] and reported here for the first time in *G. procumbens* leaves. In turn, the presence of fourteen fully identified constituents, including protocatechuic acid (**2**), *p*-hydroxybenzoic acid (**7**), neochlorogenic acid (**5**), chlorogenic acid (**10**), cryptochlorogenic acid (**12**), (−)-epicatechin (**18**), procyanidin B2 (**14**), gaultherin (**20**), hyperoside (**35**), isoquercitrin (**36**), miquelianin (**37**), guaijaverin (**39**), kaempferol glucuronide (**40**), and a flavonol aglycone quercetin (**41**) were confirmed in *G. procumbens* leaves [4,15,16]. Most of them have been previously detected also in the stems [8], fruits [9], and aerial parts [14] of eastern teaberry, as well as in the leaves and stems of other *Gaultheria* representatives [23,24,25,26]. In addition, methyl salicylate glycosides (**11** and **16**) have been found earlier in the aerial parts of *G. yunnanensis* [25,27] and *G. trichoclada* [28], and the whole plant of *Filipendula vulgaris* [17].

The next step of the phytochemical standardisation process was the quantitative profiling of polyphenols in the obtained leaf extracts, carried out by the compatible spectrophotometric and HPLC–PDA assays. As revealed, due to the abundant presence of salicylates of low reactivity in the Folin–Ciocalteu reaction, the accurate total phenolic content in the extracts is best reflected in the sum of low-molecular-weight constituents detectable by RP–HPLC (TPH) and total procyanidins estimated by the spectrophotometric *n*-butanol/HCl assay (TPA). Indeed, the TPH + TPA values for all tested extracts surpassed the TPC levels (Table 1), and the highest level of polyphenols was revealed for ME (397.9 mg/g). This extract also offered the most satisfactory recovery of individual phenolic classes from the leaf material, including salicylates, procyanidins, flavonoids, and chlorogenic acid isomers (Figure 2). What is more, the polyphenol recovery with ME was higher or comparable to that obtained for the analogous extracts from stems and fruits of *G. procumbens* [8,9] (Figure 5). Especially worthy of note was the high recovery of salicylates (Figure 2 and Figure 5), as they are the critical active markers of *Gaultheria* plants [14]. All these facts might initially indicate ME as the most promising candidate for further biological tests. Furthermore, as the leaves make up 60% of the aerial parts, next to the fruits (30%) and stems (10%), and are available on the evergreen plant practically throughout the whole growing season [1], the leaf material appears most advantageous for an industrial application.

It should also be pointed out that the present work reports the first accurate and detailed data on the levels of individual methyl salicylate glycosides and total content of salicylates (TSAL) in eastern teaberry leaves and extracts (Table 1, Figure 2). Previously [16], only the seasonal variability of the gaultherin content in the leaves was investigated, but the results were underestimated due to their conversion to salicin equivalents.

Phenolic compounds exhibit direct and indirect antioxidant and anti-inflammatory effects [29]. Although the direct mechanisms are less critical in vivo [30], they may be easily measured by simple, chemical in vitro tests [8,9]. Therefore, a set of such methods was applied in the present study for the quick optimisation of extraction solvent and selection of the extract most suitable for further biological studies in a cell-based model.

Oxidative stress and chronic inflammation are interdependent pathophysiological processes that simultaneously occur in several chronic inflammatory and oxidative disorders [31,32]. The overproduction of the prominent types of ROS, including O_2_^•−^, ^•^OH, H_2_O_2_, and HClO, is the leading cause of damage to the primary cellular macromolecules, accelerated inflammation, and activation of cell death pathways [33]. Previous studies have suggested that the cellular antioxidant activity of *Gaultheria* polyphenols might be partly related to their direct ROS-scavenging potential [14,22]. Indeed, we observed that the *G. procumbens* leaf extracts were potent direct antioxidants, and ME was the most effective (Table 2). Particularly worthy of interest is the strong scavenging potential of ME towards in vivo operating ROS, especially ^•^OH, which is the most reactive among the known free radicals in vivo, formed by the Fenton reaction between O_2_^•−^ and H_2_O_2_ in the presence of ferrous or copper ions [34]. This very destructive species reacts with the ring structure of nucleic acid, which might lead to DNA mutations and an increased risk of degenerative diseases [35]. In this context, the potent O_2_^•−^ and H_2_O_2_ scavenging activity of ME and its high FRAP value were also relevant. The high relative phenolic contents of ME might explain its elevated antioxidant capacity parameters as a significant contribution of polyphenols to the direct antioxidant effects of the leaf extracts was documented statistically (Appendix A).

In the next optimisation step, the direct anti-inflammatory potential of the extracts was verified with inhibition tests towards three pro-inflammatory enzymes, such as HYAL, LOX, and COX-2, which play a pivotal role in controlling the inflammatory cascade in the pathogenesis of many human diseases of affluence. HYAL catalyses the hydrolysis of hyaluronan, a significant extracellular matrix constituent, thereby increasing the tissue permeability and spreading inflammation [36]. LOX causes dioxygenation of polyunsaturated fatty acids in lipids to generate active eicosanoid products with pathological implications in numerous inflammatory disorders [37]. COX-2, in turn, is an enzyme responsible for the formation of prostanoids from arachidonic acid, including thromboxane and prostaglandins—the essential precursors of inflammation [38]. In our study, two extracts—ME and BE—revealed the most promising activity profile compared to positive standards and significantly inhibited the tested enzymes, especially HYAL and COX-2, and to a lesser extent also LOX (Table 3). However, some advantages might have been attributed to ME due to its superior COX-2 inhibitory effect compared to other extracts. In addition, similarly to the antioxidant activity tests, the potent relative reactivity of ME correlated with its elevated phenolic content, including high salicylate, procyanidin, and flavonoid levels (Appendix A). At this point, however, the question may arise whether the effect of *G. procumbens* leaf extracts on the pro-inflammatory enzymes results from the presence of phenolic compounds with a tanning (non-specific) denaturing effect, or is enzyme-specific. Firstly, in the correlation studies (Appendix A) we showed that low-molecular-weight procyanidins, lacking tanning effect [39], had a stronger impact on the anti-inflammatory activity of the tested extracts than highly polymerised homologs. Secondly, no inhibitory activity of the extracts on xanthin oxidase was revealed during the O_2_^•−^ scavenging test. Thus, taking these two aspects into account, it appears that the impact of *G. procumbens* extracts on the tested pro-inflammatory enzymes is specific.

Hydroalcoholic extracts are the most popular herbal drug forms in the pharmaceutical industry [11]. For this reason, and because of the potent direct antioxidant and anti-inflammatory capacity in non-cellular tests, ME was selected for further study in a model of human neutrophils obtained ex vivo from blood plasma buffy coats of healthy volunteers.

Neutrophils, also known as polymorphonuclear cells, constitute a dominant fraction of total circulating leukocytes in human blood and play a vital role in the front-line defence against harmful stimuli. Activated neutrophils synthesise and secrete a set of pro- and anti-inflammatory cytokines and chemokines, including interleukins and tumour necrosis factors [40], and release a variety of pro-inflammatory enzymes, such as proteolytic elastases or matrix metalloproteinases [41], that together are responsible for the development and progression of inflammation. In addition, the stimulated neutrophils also produce substantial amounts of ROS, including O_2_^•−^, ^•^OH, and H_2_O_2_, which act as signalling molecules and mediators of inflammation [42]. Since some of the pro-inflammatory enzymes and chemical mediators are molecular targets for treating inflammation-related disorders [40,43], neutrophils pose a useful cellular model for validating new anti-inflammatory drugs, including plant-based medications.

The present study showed that ME can significantly downregulate the pro-oxidant and pro-inflammatory functions of human neutrophils. The remarkable effectiveness of the extract was observed towards the release of IL-1β and ELA-2 from the LPS- and *f*MLP + cytochalasin B-stimulated cells. The cytokine IL-1β is a key mediator of the inflammatory response, produced by the innate immune system cells, such as neutrophils, monocytes, and macrophages, after pro-inflammatory stimuli [44]. Although IL-1β has essential homeostatic physiological functions, its overproduction implicated several pathophysiological symptoms during inflammation-related diseases, such as rheumatoid arthritis, neuropathic pain, inflammatory bowel disease, osteoarthritis, pharyngitis, vascular disorders, and periodontal diseases [44,45]. The second main target of ME–ELA-2–is the proteolytic enzyme responsible for the hydrolysis of collagen and elastin in the extracellular matrix [41]. The inhibition of this neutrophil-specific serine protease holds strong treatment effects in various inflammation-related disorders, such as respiratory diseases, Crohn’s disease, ulcerative colitis, skin inflammation, and ischemia-reperfusion injury relevant to myocardial infarction [46]. As many of these pathological conditions, caused by the over-secretion of IL-1β and ELA-2, are within the traditional phytotherapeutic indication for *Gaultheria* leaves [13], the observed effects of ME might support the medicinal use of the plant material and preparations thereof.

The effect of ME on the neutrophils’ functions could be explained by a significant contribution of gaultherin, cinnamtannin B-1, and miquelianin as the dominant polyphenols of *G. procumbens* leaves (Table 1). The accumulated research has revealed that gaultherin strongly inhibits the secretion of pro-inflammatory cytokines (IL-1β, TNF-α) and tissue remodelling enzymes (ELA-2), as well as reduces the ROS level in *f*MPL-treated neutrophils [9,14], and LPS-stimulated RAW264.7 murine macrophages [47]. Moreover, the anti-inflammatory activity of gaultherin in animal models is comparable to aspirin [22,24]. Miquelianin, then, is able to inhibit murine T-cell mutagenesis and the secretion of IL-2 and IL-4 by concanavalin A-stimulated lymph node cells [48] and downregulate the release of TNF-α, IL-1β, ELA-2, and ROS in neutrophils [14]. Similarly, cinnamtannin B1, a trimeric A-type procyanidin, inhibits the O_2_^•−^ generation and ELA-2 release by human neutrophils [14] and reduces the infiltration of the immune cells and TNF-α and IL-6 release in vivo, e.g., in a mice ear infection model [49]. Moreover, procyanidin trimers may also significantly inhibit the proliferation of concanavalin A- or LPS-induced splenocytes and secretion of IFN-γ and IL-2 [50]. Furthermore, procyanidin-rich fractions have been found to downregulate the release of pro-inflammatory cytokines, including IL-8, IL-6, and TNF-α, in a rat model of deep vein thrombosis [51].

The previous research has revealed that the aforementioned cellular effects are associated with specific molecular mechanisms, such as the activation of Nrf2 and inhibition of NF-κB pathways. For example, methyl salicylate 2-*O*-*β*-d-lactoside exerted strong anti-inflammatory activity by suppressing the IKK/NF-κB activation in a model of LPS-treated RAW264.7 murine macrophage cells [52]. Similarly, ethyl salicylate 2-*O*-*β*-d-glucoside inhibited LPS-induced activation of NF-κB in RAW264.7 cells by blocking the phosphorylation of inhibitor IκBα and p65 protein [53]. Likewise, the primary flavonoid glycoside of *G. procumbens* leaves, miquelianin, was demonstrated to upregulate HO-1 by activating the ERK-Nrf2 pathway in CD4+ T cells [54], and human hepatoma HepG2 cells [55]. Moreover, trimeric procyanidins protected PC12 cells from H_2_O_2_ by up-regulating the Nrf2/ARE pathway [56].

Therefore, the observed prominent reduction of the production of ROS, IL-1β, and ELA-2 from stimulated neutrophils, together with a significant scavenging capacity towards in vivo relevant free radicals (O_2_^•−^, ^•^OH, and H_2_O_2_), and the ability to inhibit the hyaluronan-degrading (HYAL) and arachidonic acid-converting (COX-2) enzymes, might indicate the selected extract as an effective pro-health agent, which could be successfully applied in the treatment of inflammatory disorders, according to the traditional indications of *G. procumbens* leaves [7,13].

## 4. Materials and Methods

### 4.1. Plant Material and Extraction

Leaves of *G. procumbens* L. were collected in October 2020 in the gardening centre of Ericaceae plants, Gospodarstwo Szkolkarskie Jan Cieplucha (54°44′ N, 19°18′ E), Konstantynow Lodzki (Poland). The seed origin and plant authentication were described previously [8]. The voucher specimen (KFG/HB/20001-GPRO-LEAVES) is stored in the Medicinal Plant Garden, Medical University of Lodz (Poland). Samples of the plant material were air-dried at 35 °C, powdered with an electric grinder, and sieved through a ø 0.315 mm sieve.

The leaf methanol–water (75:25, *v*/*v*) extract (ME), acetone extract, ethyl acetate extract (EAE), and *n*-butanol extract (BE) were prepared by direct reflux extraction of the powdered leaf samples of *G. procumbens* (100 g each) with the appropriate solvents. The extraction procedure was further continued according to the previously described scheme for the stems of *G. procumbens* [8]. The extraction yields were calculated per dry weight (dw) of the plant material.

### 4.2. Qualitative LC–MS/MS Analysis and Quantitative Phytochemical Profiling

The qualitative UHPLC–PDA–ESI–MS*^3^* analysis was carried out according to Michel et al. [15]. The total phenolic (TPC) and total proanthocyanidin (TPA) contents were determined by the Folin–Ciocalteu and *n*-butanol-HCl methods, respectively, as described previously [57]. Results were expressed as equivalents of gallic acid (GAE) and cyanidin chloride (CYE), respectively. All reagents were purchased from Sigma-Aldrich (St. Louis, MO, USA).

The quantitative HPLC–PDA assay was performed according to Olszewska et al. [14] using the same equipment and fully validated chromatographic procedure. Results were calculated in mg/g dw of the extracts. The levels of tentatively identified compounds were expressed as the equivalents of the active markers according to the PDA spectra as described previously [8].

### 4.3. Biological Activity Tests

#### 4.3.1. Non-Cellular In Vitro Models

The FRAP parameters were determined according to Olszewska et al. [57] and expressed in mmol of ferrous ions (Fe^2+^) produced by 1g of an analyte. The TBARS values were measured according to Matczak et al. [58] and expressed as IC_50_ values. The scavenging capacities towards DPPH, O_2_^•−^, ^•^OH, and H_2_O_2_ were evaluated according to Olszewska et al. [57], Michel et al. [15], and Marchelak et al. [59], respectively. The results were expressed as SC_50_ values. The uric acid production by the enzyme was also monitored in the O_2_^•−^ scavenging test [15] to assess the possibility of direct interaction of the tested extracts with xanthine oxidase, and no inhibitory effect on the enzyme was observed. The ability of the analytes to inhibit LOX and HYAL was evaluated according to Matczak et al. [58], and for COX-2 by the ELISA test following the manufacturer’s instructions (Cayman Chemical, Ann Arbor, MI, USA). The results were expressed as IC_50_ values. The analytes were tested at the final concentrations of 0.8–24.3 µg/mL, 0.8–11.6 µg/mL, 0.5–23.4 µg/mL, 3.5–300.0 µg/mL, 15.0–700.0 µg/mL, 2.5–120.0 µg/mL, 50–800 µg/mL, 2.5–100.0 µg/mL, and 50–1800 µg/mL for the DPPH, FRAP, TBARS, O_2_^•−^, ^•^OH, H_2_O_2_, LOX, HYAL, and COX-2 tests, respectively. Quercetin and Trolox were used as positive controls in antioxidant activity tests, while indomethacin and dexamethasone were applied in anti-inflammatory assays. All reagents and standards were purchased from Sigma-Aldrich (St. Louis, MO, USA). All non-cellular tests were performed using 96-well plates and a microplate reader SPECTROstar Nano (BMG Labtech GmbH, Ortenberg, Germany).

#### 4.3.2. Cellular Model of Human Neutrophils Ex Vivo

Neutrophils were isolated from buffy coat fractions of human blood purchased from the Warsaw Blood Donation Centre. The blood samples were collected from healthy adult human donors (18–35 years old), and routine laboratory tests showed all blood values within the normal ranges. The study conformed to the principles of the Declaration of Helsinki (the approval of the bioethics committee was not required).

The isolation was carried out with a standard method of dextran sedimentation before hypotonic lysis of erythrocytes and centrifugation in a Ficoll Hypaque gradient, according to Michel et al. [8]. As a result, the purity of the neutrophil fraction was over 97%.

The potential cytotoxicity (influence on cell wall integrity) of the analytes was investigated by flow cytometry using propidium iodide (PI) staining according to Michel et al. [8]. The results were expressed as a percentage of PI(+) cells.

The ROS levels in *f*MLP-stimulated neutrophils were determined in a luminol-dependent chemiluminescence test [8]. The release of cytokines (IL-8, IL-1β, and TNF-α) and MMP-9 by neutrophils stimulated by LPS (from *Escherichia coli*, O111:B4) were evaluated by ELISA tests following the manufacturer’s instructions (BD Biosciences, San Jose, CA, USA or R&D Systems, Minneapolis, MN, USA) as described earlier [8]. The *f*MLP + cytochalasin B-induced secretion of ELA-2 by the cells was measured using SAAVNA as a substrate, according to Michel et al. [8]. The selected extract was tested at 50–150 µg/mL. The positive controls of quercetin and dexamethasone were analysed at 25 µM. All reagents and media were purchased from Sigma-Aldrich (St. Louis, MO, USA). All cell tests were performed using 96-well plates and a microplate reader (Synergy 4, BioTek, Winooski, VT, USA).

### 4.4. Statistical and Data Analysis

The results were expressed as the means ± standard deviation (SD) for replicate determinations. The statistical analyses (calculation of SD, one-way analysis of variance, HSD Tukey tests, and linearity studies) were performed using Statistica12Pl software for Windows (StatSoft Inc., Krakow, Poland), with *p* < 0.05 regarded as significant.

## 5. Conclusions

The present study is the first detailed report on the leaf dry extracts of *G. procumbens*, prepared by direct extraction of the plant material, their polyphenolic profile, standardisation, biological effects in non-cellular and cell-based models, as well as cellular safety. The results proved that the leaves are rich in structurally and functionally diversified polyphenols, mainly methyl salicylate glycosides, free catechins, procyanidins, and flavonoids. Among the four solvents, methanol–water (75:25, *v/v*) was the best extractant for effectively recovering the active leaf polyphenols. The obtained extract (ME) revealed the strongest direct scavenging activity towards three in vivo relevant oxidants (O_2_^•−^, ^•^OH, and H_2_O_2_) and was a potent inhibitor of HYAL and COX-2 enzymes. Moreover, ME significantly and in a dose-dependent manner modulated the pro-oxidant and pro-inflammatory functions of human neutrophils ex vivo, and especially downregulated the secretion of ROS, IL-1β, and ELA-2. Therefore, the selected hydroalcoholic extract is a good candidate for a new plant-based formulation for the treatment of chronic inflammatory disorders. However, more insightful studies are still required to evaluate other molecular mechanisms of antioxidant and anti-inflammatory activity of the analysed extract and verify its in vivo effects. Up to this point, the applied standardisation scheme might be recommended for the quality control of the extract in future investigations.

## Figures and Tables

**Figure 1 molecules-27-03357-f001:**
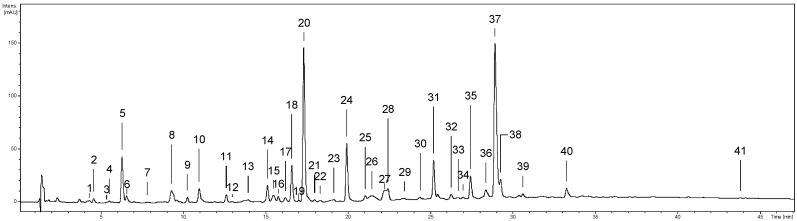
Representative UHPLC–PDA chromatograms of the methanol–water (75:25, *v*/*v*) leaf extract (ME) at 280 nm. The peak numbers refer to those implemented in Appendix A.

**Figure 2 molecules-27-03357-f002:**
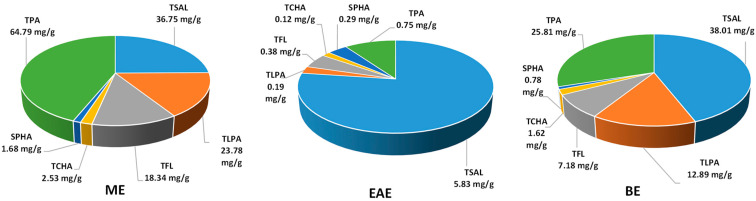
Recovery of polyphenols from the leaves with ME, EAE, and BE, expressed per leaf dw. TPA: total proanthocyanidin content (*n*-butanol/HCl assay); TSAL: total content of salicylates (HPLC); TLPA: total content of proanthocyanidins (HPLC); TFL: total content of flavonoids (HPLC); TCHA: total content of chlorogenic acid isomers (HPLC); SPHA: total content of simple hydroxybenzoic and hydroxycinnamic acids (HPLC).

**Figure 3 molecules-27-03357-f003:**
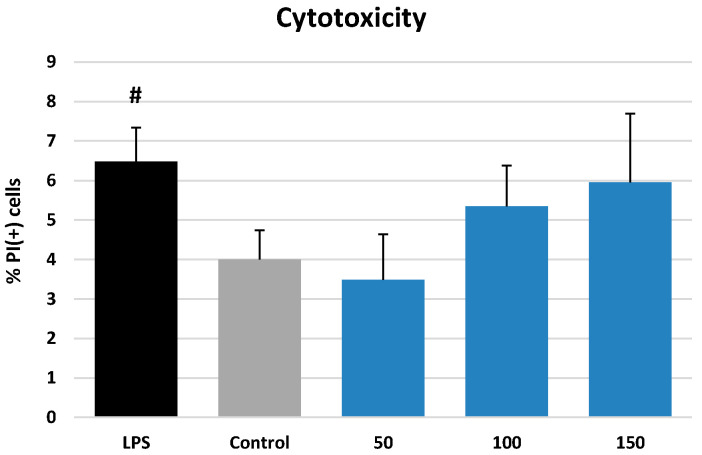
Effect of ME at 50, 100, and 150 µg/mL on viability (membrane integrity) of neutrophils as indicated by propidium iodide positive PI(+) cells. Data expressed as means ± SD of three independent experiments performed with cells isolated from five independent donors. Statistical significance: # *p* < 0.05 compared to the non-stimulated control.

**Figure 4 molecules-27-03357-f004:**
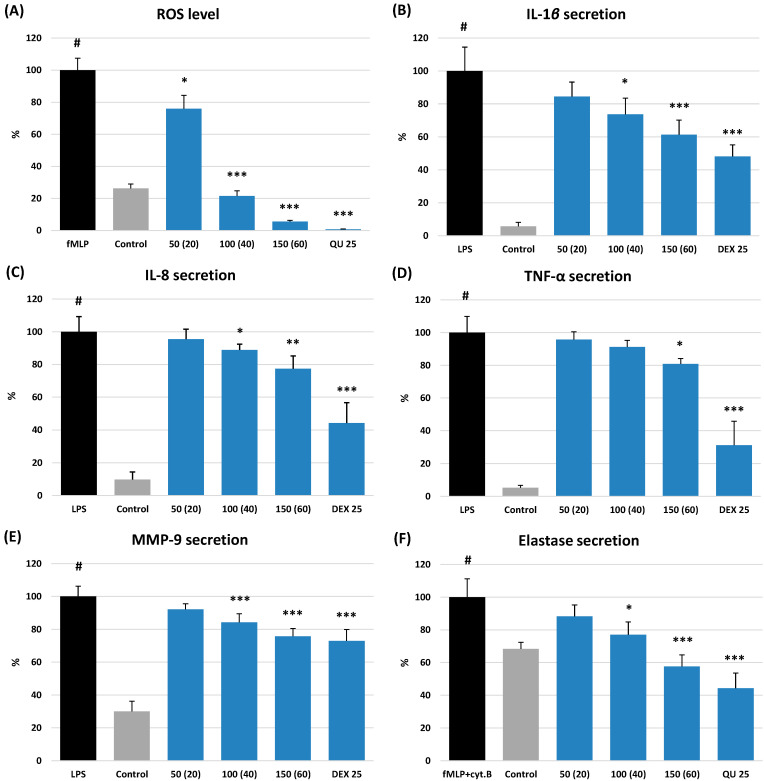
Effect of ME at 50–150 µg/mL (20–60 µg TPH + TPA/mL) on: (**A**) ROS level, and secretion of (**B**) IL-1β, (**C**) IL-8, (**D**) TNF-α, (**E**) MMP-9, and (**F**) ELA-2 by stimulated human neutrophils. Data expressed as means ± SD of three independent experiments performed with cells isolated from five independent donors. Statistical significance: # *p* < 0.001 compared to the non-stimulated control; * *p* < 0.05, ** *p* < 0.01, *** *p* < 0.001 decreased compared to the stimulated control. Positive controls tested at 25 µM: quercetin (QU) and dexamethasone (DEX).

**Figure 5 molecules-27-03357-f005:**
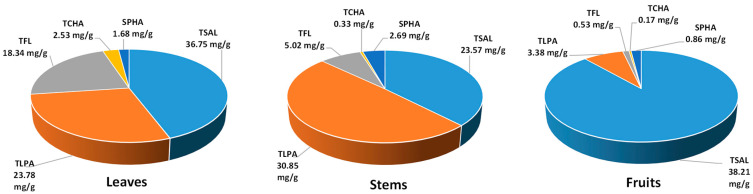
Recovery of polyphenols from the *G. procumbens* leaves, stems, and fruits with methanol–water (75:25, *v*/*v*) extracts. TSAL: total content of salicylates; TLPA: total content of proanthocyanidins; TFL: total content of flavonoids; TCHA: total content of chlorogenic acid isomers; SPHA: total content of simple hydroxybenzoic and hydroxycinnamic acids. Results expressed per dw of the plant material. Values for the stem and fruit extracts were recalculated from the previously published original data [8,9].

**Table 1 molecules-27-03357-t001:** Extraction yield and phenolic profile of *G. procumbens* leaf dry extracts (mg/g dw).

Compound/Fraction	Methanol–Water (ME)	Ethyl Acetate (EAE)	*n*-Butanol (BE)
**Extraction yield**	371.57 ± 15.57 ^C^	20.23 ± 1.01 ^A^	295.56 ± 11.78 ^B^
**Phenolic fractions:**			
TPC	302.35 ± 1.01 ^C^	168.12 ± 0.69 ^A^	174.93 ± 1.64 ^B^
TPH	223.54 ± 3.02 ^B^	336.72 ± 4.04 ^C^	204.63 ± 3.66 ^A^
TSAL	98.89 ± 0.47 ^A^	288.13 ± 3.91 ^C^	128.61 ± 3.06 ^B^
TPA	174.38 ± 2.35 ^C^	36.93 ± 0.35 ^A^	87.32 ± 1.37 ^B^
TLPA	63.99 ± 2.01 ^C^	9.67 ± 0.16 ^A^	43.64 ± 1.14 ^B^
TFL	49.35 ± 0.69 ^C^	18.59 ± 0.27 ^A^	24.28 ± 0.19 ^B^
TPHA	11.31 ± 0.13 ^B^	20.33 ± 0.12 ^C^	8.10 ± 0.12 ^A^
TCHA	6.80 ± 0.09 ^C^	5.88 ± 0.08 ^B^	5.47 ± 0.08 ^A^
SPHA	4.51 ± 0.07 ^B^	14.45 ± 0.16 ^C^	2.63 ± 0.06 ^A^
**Primary compounds:**			
Neochlorogenic acid (**5**)	4.24 ± 0.08 ^B^	2.89 ± 0.03 ^A^	2.91 ± 0.09 ^A^
Chlorogenic acid (**10**)	1.68 ± 0.04 ^B^	2.99 ± 0.04 ^C^	1.31 ± 0.05 ^A^
Cryptochlorogenic acid (**12**)	0.88 ± 0.04 ^A^	*n.d.*	1.25 ± 0.02 ^B^
Methyl salicylate triglycoside (**11**)	0.49 ± 0.01 ^A^	*n.d.*	0.79 ± 0.04 ^B^
Gaultherin (**20**)	98.41 ± 0.31 ^A^	288.13 ± 3.91 ^C^	127.81 ± 2.81 ^B^
(-)-Epicatechin (**18**)	9.07 ± 0.29 ^C^	7.91 ± 0.13 ^B^	0.54 ± 0.02 ^A^
Procyanidin B2 (**14**)	13.15 ± 0.37 ^B^	*n.d.*	6.29 ± 0.08 ^A^
Cinnamtanin B-1 (**24**)	18.61 ± 0.28 ^C^	0.77 ± 0.03 ^A^	12.89 ± 0.58 ^B^
Wintergreenoside A (**31**)	7.26 ± 0.24 ^B^	*n.d.*	1.24 ± 0.04 ^A^
Hyperoside (**35**)	5.01 ± 0.16 ^A^	9.46 ± 0.37 ^C^	5.33 ± 0.26 ^B^
Isoquercitrin (**36**)	1.16 ± 0.05 ^B^	2.97 ± 0.09 ^C^	1.10 ± 0.04 ^A^
Miquelianin (**37**)	32.14 ± 0.69 ^C^	1.55 ± 0.06 ^A^	6.72 ± 0.13 ^B^
Guaijaverin (**39**)	1.46 ± 0.02 ^A^	4.17 ± 0.19 ^C^	1.73 ± 0.07 ^B^
Quercetin (**41**)	0.22 ± 0.01 ^A^	0.43 ± 0.02 ^B^	7.07 ± 0.17 ^C^

Results are presented as means ± SD (*n* = 3). Means with different superscript capital letters (A–C) within the same row differ significantly (*p* < 0.05). Extraction yield calculated per dry weight (dw) of leaves and other results per dw of the extracts. TPC: total phenolic content (Folin–Ciocalteau assay) in gallic acid equivalents (GAE); TPH: total phenolic content (HPLC); TSAL: total content of salicylates (HPLC); TPA: total content of proanthocyanidins (*n*-butanol/HCl assay) in cyanidin chloride equivalents (CyE); TLPA: total content of proanthocyanidins (HPLC); TFL: total content of flavonoids (HPLC); TPHA: total content of phenolic acids (HPLC); TCHA: total content of chlorogenic acid isomers (HPLC); SPHA: total content of simple hydroxybenzoic and hydroxycinnamic acids (HPLC). Numbers in parentheses refer to the peak numbering in Figure 1 and Appendix A. *n.d.*—not detected.

**Table 2 molecules-27-03357-t002:** Antioxidant activity of *G. procumbens* leaf dry extracts (ME, EAE, and BE) in non-cellular models.

Analyte	DPPH	FRAP	TBARS	O_2_^•−^	^•^OH	H_2_O_2_
	SC_50_ (µg/mL) ^a^	mmol Fe^2+^/g ^b^	IC_50_ (µg/mL) ^c^	SC_50_ (µg/mL) ^a^	SC_50_ (µg/mL) ^a^	SC_50_ (µg/mL) ^a^
**ME**	6.77 ± 0.28 ^C^	6.36 ± 0.14 ^C^	8.46 ± 0.19 ^C^	26.33 ± 0.88 ^B^	152.04 ± 4.28 ^B^	44.41 ± 1.96 ^C^
**EAE**	14.17 ± 0.29 ^E^	3.82 ± 0.13 ^A^	14.71 ± 2.06 ^E^	39.30 ± 1.55 ^C^	480.77 ± 13.01 ^E^	83.32 ± 2.98 ^D^
**BE**	8.33 ± 0.09 ^D^	4.41 ± 0.18 ^B^	10.68 ± 0.46 ^D^	62.36 ± 2.31 ^D^	236.51 ± 8.64 ^D^	43.25 ± 1.61 ^C^
**QU**	1.52 ± 0.03 ^A^	49.04 ± 0.59 ^F^	1.69 ± 0.05 ^A^	7.35 ± 0.19 ^A^	41.07 ± 3.89 ^A^	6.96 ± 0.42 ^A^
**TX**	4.23 ± 0.05 ^B^	12.56 ± 0.23 ^D^	4.58 ± 0.33 ^B^	142.15 ± 2.19 ^E^	172.26 ± 3.01 ^C^	15.76 ± 0.31 ^B^

^a^ SC_50_: scavenging efficiency in μg of the dry extract or compound per mL of the reaction solution; ^b^ antioxidant activity expressed in mmol of ferrous ions (Fe^2+^) produced by 1 g of the dry extract or standard; ^c^ IC_50_: inhibition concentration in µg of the dry extract or compound per mL of the reaction solution. The positive controls: QU (quercetin) and TX (Trolox^®^, (±)-6-hydroxy-2,2,7,8-tetramethyl- chroman-2-carboxylic acid). Results presented as mean values ± SD (*n* = 3). For each parameter different superscript capital letters (A–E) indicate significant differences (*p* < 0.05).

**Table 3 molecules-27-03357-t003:** Inhibitory activity of *G. procumbens* leaf dry extracts (ME, EAE, and BE) towards pro-inflammatory enzymes.

Analyte	HYAL	LOX	COX-2
	IC_50_ (µg/mL) ^a^	IC_50_ (µg/mL) ^a^	IC_50_ (μg/mL) ^a^
**ME**	18.66 ± 0.86 ^C^	351.55 ± 13.76 ^D^	711.08 ± 25.55 ^D^
**EAE**	34.57 ± 1.82 ^D^	626.25 ± 11.19 ^E^	1416.93 ± 50.85 ^F^
**BE**	14.63 ± 0.66 ^B^	267.04 ± 12.98 ^C^	970.64 ± 38.53 ^E^
**IND**	12.68 ± 1.79 ^A^	91.89 ± 3.95 ^A^	184.32 ± 8.56 ^A^
**DEX**	14.07 ± 1.25 ^B^	120.16 ± 4.86 ^B^	511.23 ± 14.58 ^C^
**QU**	31.78 ± 1.52 ^D^	88.35 ± 7.37 ^A^	469.46 ± 14.52 ^B^

^a^ IC_50_: inhibition concentration in μg of the dry extract or compound per mL of the reaction solution. The positive controls: IND (indomethacin), DEX (dexamethasone), and QU (quercetin). Results are presented as mean values ± SD (*n* = 3). For each parameter different superscript capital letters (A–F) indicate significant differences (*p* < 0.05).

## Data Availability

Not applicable.

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
