# Peer review of "The Effect of Standardised Leaf Extracts of Gaultheria procumbens on Multiple Oxidants, Inflammation-Related Enzymes, and Pro-Oxidant and Pro-Inflammatory Functions of Human Neutrophils"

_molecules, 2022, doi:10.3390/molecules27103357_

Round 1

Reviewer 1 Report

The paper by Piotr Michel et al describes the antioxidant and anti-inflammatory activities of an extract from the leaves of Gaultheria procumbens.

The paper is clear and well written, and the topic is suitable to the journal, however I have some major comments on experiments and discussion.

Why methanol and not ethanol was used for the leaves extraction? EtOH is safe and can be obtained by green chemistry and these aspects should be considered for an industrial development.

It is not clear to me if this is the first paper reporting a qualitative and quanti composition of polyphenols from the leaves of the plant. If this is true, this aspect should be emphasized , if not, a comparison with data from the literature should be given (a Venn diagram could be useful).

The effects of the extract on the inflammatory enzymes is found to be related, besides to salycilate content, even to proanthocyanidins which have a well known tanning effect. Is the effect on the tested enzymes specific or it is a result of a tanning (non-specific) denaturating effect?. This should be investigated by measuring the tanning effect of the extracts or testing the enzyme inhibiting effects on some other enzymes not involved in the inflammatory cascade.

Based on thermodynamic and kinetic studies, it is now well established that low molecular compound are far to act by scavenging radicals and peroxides. Most of the low molecular weight antioxidant compounds acts by activating the Nrf2 nuclear factor which has a pivotal role in the neutrophils activation. Most f the polyphenols identified in the extract have a catechol moiety and are potentially able to act as Nrf2 activators. This aspect should be carefully considered in the experiments and discussion.

Fig 4 the concentration of QU and Dex should be reported in the legend.

Author Response

Authors’ answers to the Reviewers’ remarks:

With regard to the manuscript Ref. No.molecules-1725632

Michel et al.: „The Effect of Standardised Leaf Extracts of Gaultheria procumbens L. on Multiple Oxidants, Inflammation-Related Enzymes, and Pro-oxidant and Pro-inflammatory Functions of Human Neutrophils”.

Hereby, we would like to express our gratitude to the Reviewers for all their valuable remarks that allowed us to improve the manuscript and hopefully present it in a form suitable for publication in the Molecules.

Reviewer 1:

  1. “Why methanol and not ethanol was used for the leaves extraction? EtOH is safe and can be obtained by green chemistry and these aspects should be considered for an industrial development.”

According to the Reviewer's question, we would like to explain that initially, methanol was indeed considered a non-green solvent, but this trend has changed in recent years. Advanced developments in the chemical industry and the use of new production procedures have positively impacted the cost, safety, and health issues related to methanol. Nowadays, the amount of energy necessary for the production, recycling, and treatment of methanol has decreased. A suitable screening indicator to illustrate these trends is the cumulative energy demand (CED) [Capello et al., 2007, Green Chem., 9, 927-934]. For example, for ethanol, CED (distillation) is 18,9 per kg solvent/MJ-eq. and CED (incineration) is 18,4 per kg solvent/MJ-eq. In the case of methanol CED (distillation) is 19,0 per kg solvent/MJ-eq. and CED (incineration) is 18,5 per kg solvent/MJ-eq. These similar results for methanol and ethanol indicate both as environmentally friendly solvents [Capello et al., 2007, Green Chem., 9, 927-934; Henderson et al., 2011, Green Chem., 13, 854-862; Prat et al., 2014, Green Chem., 16, 4546; Płotka-Wasylka et al., 2017, Trends Anal. Chem., 91, 12-25]. In addition, methanol is much cheaper than ethanol. Even though ethanol is generally preferred for some applications, methanol, and especially its aqueous solutions, are also widely used, for instance in the analysis of plant extracts and for the production of dry extracts for medicinal purposes [Truong et al., 2019, J. Food Qual., ID 8178294; Shahidi, 2022, Food Chem. Toxicol., 163, ID 112981; Makhija et al., 2022, J. Ethnopharmacol., 287, ID 114953; Taghavi et al., 2022, Foods, 11(8), ID 1072; Feng, 2022, Molecules, 27(7), ID 2268].

  1. “It is not clear to me if this is the first paper reporting a qualitative and quanti composition of polyphenols from the leaves of the plant. If this is true, this aspect should be emphasized, if not, a comparison with data from the literature should be given (a Venn diagram could be useful).”

Taking into account the Reviewer’s concerns about the novelty of the research on the qualitative and quantitative composition of G. procumbens leaf polyphenols, we would like to confirm that the present studies were performed for the first time on the dry extracts prepared by direct extraction of the leaves with four various solvents. On the other hand, our previous research on the leaf extracts, which the Reviewer is likely to have in mind, concerned fractionated leaf dry extracts obtained by sequential liquid-liquid extraction with organic solvents of different polarities [Michel et al., 2014, Molecules, 19, 20498-20520]. As the chemical composition of plant extracts may vary greatly depending on the extraction solvent, extraction procedure, and due to the seasonal variability of the plant material, which in turn is influenced by several climatic and geographical factors [Isah, 2019, Biol. Res., 52(1), 39] (Ref. nr 22), the thorough phytochemical standardization of the newly prepared extracts was required. Moreover, the chemical composition of the extracts and plant material in our previous article was insufficiently recognized, mainly due to the lack of available phenolic standards. Consequently, regardless of the previous paper, significant novel information on the phytochemistry of G. procumbens leaves was given in the present submission. The novelty concerning the qualitative and quantitative composition of leaf polyphenols in this work has been highlighted in different sections of the previous manuscript version (lines 278-280, 282-287, 317-321, 525-527). However, we agree with the Reviewer that this aspect might be improved, therefore, to clarify it for the Readers, this issue was now also emphasized in other sections (lines 272-277).

  1. “The effects of the extract on the inflammatory enzymes is found to be related, besides to salicylate content, even to proanthocyanidins which have a well-known tanning effect. Is the effect on the tested enzymes specific or it is a result of a tanning (non-specific) denaturating effect?. This should be investigated by measuring the tanning effect of the extracts or testing the enzyme inhibiting effects on some other enzymes not involved in the inflammatory cascade.”

The presence of highly polymerized proanthocyanidins has been found previously in stems and fruits of G. procumbens [Michel et al., 2019, Int. J. Mol. Sci., 20, 1-17; Michel et al., 2020, Food Funct., 11, 7532-7544], and was also pointed out in the present manuscript for the leaves (lines 102, 128-133). We agree with the Reviewer’s suggestion, that some phenolic compounds may contribute to the biological activity of plant extracts through protein binding, which for the condensed compounds may mean a tanning effect. Indeed, this is the basis of the well-known anti-inflammatory effects of tannins [Sieniawska, 2015, Nat. Prod. Commun., 10(11), 1877-1884] (Ref. nr 39). However, in the present study, we showed that low-molecular-weight proanthocyanidins (TLPA: total content of proanthocyanidins by HPLC method), lacking a tanning effect, have a stronger impact on the antioxidant and anti-inflammatory activity of the studied extracts than the highly polymerized homologs, reflected in the TPA values (TPA: total content of proanthocyanidins by n-butanol/HCl spectrophotometric assay, understood as the sum of low-molecular-weight and highly condensed proanthocyanidins). This fact was confirmed by higher correlation coefficients observed between the IC50 values in the enzyme inhibition assays and TLPA (  0.82) values than for the TPA levels (  0.62), which was also indicated in the manuscript (lines 185-189). At the same time, we would like to explain that we have investigated not only the direct effect of the extracts on pro-inflammatory enzymes, but also on xanthine oxidase. This issue was studied during the O2•‒ scavenging test. The reaction of xanthine (substrate) with xanthine oxidase produces uric acid, the level of which is measured directly spectrophotometrically at λ = 295 nm. At the same time, a by-product of this reaction is O2•‒. The ability to react directly with O2•‒ (generated in the test environment by xanthine oxidase) was confirmed for the extracts by no inhibitory effect on xanthine oxidase, which was in turn evidenced by the normal production of uric acid. Thus, taking these two aspects into account, it has been shown that the effect of the G. procumbens extracts on the pro-inflammatory enzymes is specific rather than derived from the denaturing activity of tannin-like polyphenolic compounds. This question was addressed in the Results (lines 191-194) and Materials and Methods (lines 482-485). To clarify this for the Readers, the appropriate changes were also introduced into the Discussion section of the manuscript (lines 367-377).

  1. “Based on thermodynamic and kinetic studies, it is now well established that low molecular compounds are far to act by scavenging radicals and peroxides. Most of the low molecular weight antioxidant compounds acts by activating the Nrf2 nuclear factor which has a pivotal role in the neutrophils activation. Most of the polyphenols identified in the extract have a catechol moiety and are potentially able to act as Nrf2 activators. This aspect should be carefully considered in the experiments and discussion.”

We agree with the Reviewer, that the direct mechanism of antioxidant and anti-inflammatory activity in vivo is not dominant for polyphenols, which we also noted in the article (line 330). However, non-cellular methods of measuring antioxidant and anti-inflammatory activities are simple, fast, sensitive, and low-cost and therefore are often used for phytochemical screening, which is also applied in the presented work for the quick extraction solvent optimization. This fact was emphasized in the original version of the manuscript (lines 331-333). Concerning the indirect molecular mechanisms of antioxidant and anti-inflammatory activity of polyphenols, they are well-documented in the literature, and polyphenols are known to act as activators of Nuclear factor erythroid 2-related factor 2 (Nrf2) and inhibitors of the MAPK/NF-κB signaling pathways [Scapagnini et al., 2011, Mol. Neurobiol., 44(2), 192-201; Hussain et al., 2017, Oxid. Med. Cell. Longev., ID 8254289; Karunaweera et al., 2015, Front. Mol. Neurosci., 8, 24; Khan et al., 2020, Crit. Rev. Food Sci. Nutr., 60(16), 2790-2800]. These biological activity mechanisms were also confirmed for methyl salicylate glycosides isolated from various Gaultheria members. For example, methyl salicylate 2-O-β-D-lactoside, isolated previously from Gaultheria yunnanensis, exerted strong anti-inflammatory activity by suppressing the IKK/NF-κB activation in a model of LPS-treated RAW264.7 murine macrophage cells [Zhang et al., 2012, Mol. Pharmaceutics, 9, 671-677]. Similarly, ethyl salicylate 2-O-β-D-glucoside (also obtained from Gaultheria yunnanensis) inhibited LPS-induced activation of NF-κB in RAW264.7 cells by blocking phosphorylation of inhibitor IκBα and p65 protein [Xin et al., 2013, Int. Immunopharmacol., 15, 303-308]. Likewise, the primary flavonoid glycoside of G. procumbens leaves, miquelianin, was demonstrated to upregulate HO-1 by activating the ERK-Nrf2 pathway in CD4+ T cells [Choi et al., 2021, Antioxidants, 10, 1120] and human hepatoma HepG2 cells [Lee et al., 2018, Food Sci. Biotechnol., 27, 809-817]. Moreover, trimeric procyanidins protected PC12 cells from H2O2 by up-regulating the Nrf2/ARE pathway [Chen et al., 2022, Molecules, 27, 2308].

Summing up, as indicated above, the activation of Nrf2 and inhibition of NF-κB pathways were already demonstrated for dominant phenolic compounds of G. procumbens leaves. On the other hand, the inflammatory process is controlled by numerous mechanisms; therefore, activating Nrf2 does not necessarily tell us everything about the effects of the process, including the release of various pro-inflammatory factors, such as ROS or pro-inflammatory cytokines. Consequently, as some of such pro-inflammatory mediators are molecular targets for treating inflammation-related disorders [Jones et al., 2016, Semin. Immunol., 28, 137-145; Garin and Proudfoot, 2011, Exp. Cell Res., 317, 602-612], it is so important to examine their secretion by the innate immune system cells. Therefore, our goal was to measure the influence of our extracts on the generation of pro-oxidant (ROS level) and pro-inflammatory factors [level of cytokines: IL-1β, IL-8, and TNF-α, and tissue remodeling enzymes, such as matrix metalloproteinase 9 (MMP-9) and elastase-2 (ELA-2)] by human neutrophils ex vivo (lines 86-90), and not to investigate the basic molecular mechanisms of these processes because they may be anticipated from the previous works.

To clarify this for the Readers, the necessary changes have been implemented in the Discussion section (lines 430-440), and the added literature was highlighted in red in the References section (Ref. numbers 52-56) in the revised version of the manuscript. We sincerely hope that the Reviewer will find all these changes appropriate and that the submission in its present form will be suitable for publication. We would also like to ensure, that more insightful studies evaluating molecular mechanisms of antioxidant and anti-inflammatory activity of the analyzed extracts, for instance, inhibition of Toll-Like Receptor 4 (TLR4) activation or inhibition of Intercellular Adhesion Molecule 1 (ICAM-1) and Vascular Cell Adhesion Molecule 1 (VCAM-1) expression in various human cells will be included in our future research, because such molecular studies would require novel experimental plan.

  1. “Fig 4 the concentration of QU and Dex should be reported in the legend.”

As to the Reviewer’s suggestion, we would like to clarify that the concentration of QU and DEX was given in the legend under Fig. 4 in the original version of the manuscript (lines 230-231).

Reviewer 2 Report

This paper deals with antioxidant and anti-inflammatory effects of standardised leaf extracts.

Aim of the study was to optimize of the solvent for the effective recovery of active leaf components through simple direct extraction and to verify the biological effects of the selected extract.

The work is part of a research line of the author group, who for years worked on, on the phytochemical profiling of different extracts of Gaultheria and their biological activity. The results demonstrate a satisfactory recovery of phenolic compounds from extracts obtained by direct extraction of dried biomass, and specifically that the methanol-water extract shows the most satisfactory recovery of individual phenolic and could be then considered as candidate for a new plant-based formulation for the treatment of chronic inflammatory disorders.

The work adds new insights in the study of Gaultheria extracts, despite in vivo studies on the biological activity of the extracts are still lacking. In my opinion this is no reason to prevent publication of the work, since in vivo studies require a lengthy investigation which could be the subject of subsequent work.

The manuscript appears well-defined and exhaustive, and it seems qualifying for the journal.

I think that the description of the methodology is exhaustive, as well as is the presentation of the results and the basic statistics applied.

The conclusions address the posed questions.

References are presented according with the Journal guidelines, and I do not have additional comments on tables and figures.

My only suggestion is to delete the author mane (L.) from the plant name in the title, and in my opinion this paper is acceptable as it is

Author Response

Authors’ answers to the Reviewers’ remarks:

With regard to the manuscript Ref. No.molecules-1725632

Michel et al.: „The Effect of Standardised Leaf Extracts of Gaultheria procumbens L. on Multiple Oxidants, Inflammation-Related Enzymes, and Pro-oxidant and Pro-inflammatory Functions of Human Neutrophils”.

Hereby, we would like to express our gratitude to the Reviewers for all their valuable remarks that allowed us to improve the manuscript and hopefully present it in a form suitable for publication in the Molecules.

Reviewer 2:

  1. “My only suggestion is to delete the author name (L.) from the plant name in the title, and in my opinion this paper is acceptable as it is.”

We would like to thank the Reviewer for the positive feedback. According to the Reviewer’s suggestion, the letter “L.” was removed from the title of the article and supplementary materials.

Round 2

Reviewer 1 Report

The comments have been suitable replied